# In Silico and In Vitro Evaluation of Some Amidine Derivatives as Hit Compounds towards Development of Inhibitors against Coronavirus Diseases

**DOI:** 10.3390/v15051171

**Published:** 2023-05-15

**Authors:** Ahmed H. E. Hassan, Selwan M. El-Sayed, Mizuki Yamamoto, Jin Gohda, Takehisa Matsumoto, Mikako Shirouzu, Jun-ichiro Inoue, Yasushi Kawaguchi, Reem M. A. Mansour, Abtin Anvari, Abdelbasset A. Farahat

**Affiliations:** 1Department of Medicinal Chemistry, Faculty of Pharmacy, Mansoura University, Mansoura 35516, Egypt; salwanmahmoud@mans.edu.eg (S.M.E.-S.); reemmansour@std.mans.edu.eg (R.M.A.M.); 2Research Center for Asian Infectious Diseases, Institute of Medical Science, The University of Tokyo, Tokyo 108-8639, Japan; mizuyama@g.ecc.u-tokyo.ac.jp (M.Y.); jgohda@g.ecc.u-tokyo.ac.jp (J.G.); ykawagu@g.ecc.u-tokyo.ac.jp (Y.K.); 3Drug Discovery Structural Biology Platform Unit, RIKEN Center for Biosystems Dynamics Research, Kanagawa 230-0045, Japan; takehisa.matsumoto@riken.jp (T.M.); mikako.shirouzu@riken.jp (M.S.); 4Infection and Advanced Research Center (UTOPIA), The University of Tokyo Pandemic Preparedness, Tokyo 108-8639, Japan; jun-i@g.ecc.u-tokyo.ac.jp; 5Division of Molecular Virology, Department of Microbiology and Immunology, The Institute of Medical Science, The University of Tokyo, Tokyo 108-8639, Japan; 6Master of Pharmaceutical Sciences Program, California Northstate University, 9700 W Taron Dr., Elk Grove, CA 95757, USA; abtin.anvari8815@cnsu.edu; 7Department of Pharmaceutical Organic Chemistry, Faculty of Pharmacy, Mansoura University, Mansoura 35516, Egypt

**Keywords:** antiviral agents, coronaviruses, viral entry

## Abstract

Coronaviruses, including SARS-CoV-2, SARS-CoV, MERS-CoV and influenza A virus, require the host proteases to mediate viral entry into cells. Rather than targeting the continuously mutating viral proteins, targeting the conserved host-based entry mechanism could offer advantages. Nafamostat and camostat were discovered as covalent inhibitors of TMPRSS2 protease involved in viral entry. To circumvent their limitations, a reversible inhibitor might be required. Considering nafamostat structure and using pentamidine as a starting point, a small set of structurally diverse rigid analogues were designed and evaluated in silico to guide selection of compounds to be prepared for biological evaluation. Based on the results of in silico study, six compounds were prepared and evaluated in vitro. At the enzyme level, compounds **10**–**12** triggered potential TMPRSS2 inhibition with low micromolar IC_50_ concentrations, but they were less effective in cellular assays. Meanwhile, compound **14** did not trigger potential TMPRSS2 inhibition at the enzyme level, but it showed potential cellular activity regarding inhibition of membrane fusion with a low micromolar IC_50_ value of 10.87 µM, suggesting its action could be mediated by another molecular target. Furthermore, in vitro evaluation showed that compound **14** inhibited pseudovirus entry as well as thrombin and factor Xa. Together, this study presents compound **14** as a hit compound that might serve as a starting point for developing potential viral entry inhibitors with possible application against coronaviruses.

## 1. Introduction

Coronaviruses (CoVs) are a family of viruses known to cause respiratory, enteric, and neurogenic diseases. To date, seven human coronaviruses (hCoVs) are known. The outbreaks of severe acute respiratory syndrome (SARS-CoV) in 2002, Middle East respiratory syndrome (MERS-CoV) in 2012, and COVID-19 (SARS-CoV-2) by the end of 2019 highlight the serious threats of coronaviruses infections. Health emergencies were declared as a result of epidemics and/or pandemics caused by these viruses. SARS-CoV emerged in China and spread to 29 countries, while MERS-CoV was identified in Jordan and spread in 27 countries with 80% of the reported cases in Saudi Arabia [1,2]. In addition, the new SARS-CoV-2 that emerged in China and causes COVID-19 has spread globally. As of March 2020, the World Health Organization (WHO) declared COVID-19 as a global pandemic. It resulted in more than 660 million reported COVID-19 cases and more than six million reported deaths. The real number is estimated to be much higher as many cases were not documented. Despite devising vaccines against SARS-CoV-2, they do not provide enough protection but reduce the risk of developing severe symptoms. Furthermore, such vaccines become less effective against the emerging variants as SARS-CoV-2 continuously mutates. Therefore, even vaccinated people are still prone to catch infections especially when exposed to the emerging viral mutant variants such as omicron, delta and other variants harboring diverse mutations particularly within spike (s) proteins [3,4,5]. Unfortunately, at least 65 million people suffer long COVID which is a post-COVID syndrome characterized by the persistence of SARS-CoV-2 symptoms for weeks or even months after catching infection [6,7,8]. Despite some therapeutic agents targeting viral proteins having been developed, such as those targeting main protease, resistant strains of the continuously mutating virus might evolve [9,10]. To circumvent these insufficiencies of the currently available therapeutics targeting the viral proteins that are subjected to mutations, it might be beneficial to address novel agents relying instead on host-based processes required or exploited for viral infection [11,12,13].

In principle, infection by coronaviruses including SARS-CoV, MERS-CoV, SARS-CoV-2, and influenza A viruses involves a common initial process of viral entry into the cell that starts with the binding of the viral S protein to the angiotensin-converting enzyme-2 (ACE2) receptor on the host cells. Subsequently, proteolysis of the S protein by the action of host proteases leads to viral fusion with the cell and entry of the viral RNA-encoded genome. In this process, transmembrane protease serine-2 (TMPRSS2) plays a crucial role [14]. Only subsequent to entry, can the virus recruit normal cellular functions for production of its proteins and genome to replicate. Accordingly, developing inhibitors of the fusion and entry steps would protect against new cell infections and would stall the propagation and replication of the virus. Intercepting such a conserved host-dependent process could offer more advantages than targeting the continuously mutating viral proteins. In lieu, inhibitors of the proteolytic activity of TMPRSS2 could be promising inhibitors of viral fusion and entry, and could be promising host-based targeted therapy against coronaviruses including SARS-CoV, MERS-CoV, SARS-CoV-2, and influenza A viruses. Such agents would be expected to circumvent limitation of viral resistance because of viral mutations [15,16,17,18].

## 2. Materials and Methods

### 2.1. In Silico Docking Study

A molecular docking study was performed using the crystal structure of TMPRSS2 deposited in Protein Data Bank (PDB: 7MEQ) employing standard computational protocols as described in Appendix A [19,20,21,22,23,24,25,26,27].

### 2.2. Chemistry

Compounds **5**, **11**, and **12** were reported previously [28,29]. Compounds **8**–**10** were prepared following well-established chemical reactions as detailed in the Appendix A.

### 2.3. Cells and Materials

A pair of previously described 293FT-based reporter cell lines that constitutively express individual split reporters (DSP1-7 and DSP8-11 proteins, Appendix A) were maintained in Dulbecco’s modified Eagle’s medium (DMEM) containing 10% fetal bovine serum (FBS) and 1 μg/mL puromycin. TMPRSS2 expressing VeroE6 cells (VeroE6-TMPRSS2) were maintained in Eagle’s minimum essential medium (EMEM) containing 15% fetal bovine serum (FBS) as described previously [30]. See Appendix A for recombinant enzymes and substrates used in enzyme assays.

### 2.4. Cell Fusion Assay Using Dual Split Proteins (DSP)

As shown in Appendix A, a DSP-assay was employed to perform a SARS-CoV-2 S protein/TMPRSS2/ACE2 assay for a quantitative investigation of ACE2/TMPRSS2-dependant SARS-CoV-2 S protein-mediated membrane fusion (S2TA assay). Co-transfection assay (CoTF assay) was also applied to investigate TMPRSS2-independent inhibition of membrane fusion (Appendix A). All tests were performed according to the reported protocol [31]. Briefly, for the S2TA assay, effector cells expressing S protein with DSP8-11 and target cells expressing ACE2 and TMPRSS2 with DSP1-7 were seeded in 10 cm plates and incubated overnight. Cells were treated with 6 μM EnduRen (Promega, WI, USA), a substrate for Renilla luciferase (RL), for 2 h. To test the effect of the inhibitor, 1 μL of the compound dissolved in DMSO was added to the 384-well plates (Greiner Bioscience, Frickenhausen, Germany). Next, 50 μL of each single cell suspension (effector and target cells) was added to the 384-well plates using a Multidrop dispenser (Thermo Fisher Scientific, MA, USA). After incubation at 37 °C in 5% CO_2_ for 4 h, RL activity was measured using a Centro xS960 luminometer (Berthold, Bad Wildbad, Germany). For the CoTF assay, cells expressing DSP1-7 and DSP8-11 were used to evaluate the effect of compounds on TMPRSS2-independent inhibition of cell fusion or reduction in measured luminescence and fluorescence that might arise from effect on DSP reassociation.

### 2.5. Enzyme Assays

In these assays, TMPRSS2 and thrombin were used at final concentrations of 0.3 and 3 nM, respectively. Factor Xa was used at a final concentration of 3 or 12 nM. Peptides with a fluorescence quenching pair (Dabcyl and Edans) at both ends were used as substrates for each enzyme (see Appendix A). To test the effect of compounds on enzyme activity, enzyme and compound were mixed in 90 μL of assay buffer (20 mM Tris-HCl (pH 8.0), 150 mM NaCl) at 1.25 times the final concentration in a 96-well plate. After incubation for 30 min at room temperature, 80 μL of the mixture was added to 20 μL of 50 μM substrate solution prepared from the assay buffer. Fluorescence was read every 5 min at room temperature using CLARIOstar Plus (BMG LABTECH, Ortenberg, Germany) at an excitation wavelength of 340 nm and an emission wavelength of 490 nm. The IC_50_ values of the compound were calculated using Prism version 9.0 (GraphPad Software, CA, USA).

### 2.6. Pseudovirus Assay

To produce replication-deficient vesicular stomatitis virus (VSV), BHK cells expressing T7 RNA polymerase were transfected with T7 promoter-driven expression plasmids for VSV proteins (pBS-N/pBS-P/pBS-L/pBS-G) and pΔG-Luci (a plasmid encoding VSV genomic RNA lacking the G gene and encoding firefly luciferase) as described previously [32,33]. At 48 h post-transfection, the supernatants were harvested. The 293T cells were then transfected with an expression plasmid for S protein or VSV G using calcium phosphate precipitation. At 16 h post-transfection, cells were infected with replication-deficient VSV at a multiplicity of infection (MOI) of 1. At 2 h post-infection, cells were washed and incubated for another 16 h before supernatants containing pseudovirus were harvested. For the infection assay, VeroE6-TMPRSS2 cells were seeded in 96-well plates (2 × 10^4^ cells/well) and incubated overnight. Cells were pretreated with inhibitors for 1 h prior to pseudovirus infection. Luciferase activity was measured 16 h after infection using the Bright-Glo Luciferase Assay System or ONE-Glo Luciferase Assay System (Promega) and the Centro xS960 luminometer (Berthold, Bad Wildbad, Germany).

### 2.7. Cell Toxicity Assay

To test the toxicity of the compounds, VeroE6-TMPRSS2 cells were treated with the compounds and cell viability was analyzed using the Celltiter-Glo luminescent cell viability assay (G7570, Promega) 24 h after treatment according to the manufacturer’s protocol.

## 3. Results

### 3.1. Design Rational

Hit discovery and confirmation is an indispensable first step in early drug discovery [34,35,36,37,38]. While a hit compound would show the desired type of activity, its activity might be of low potential and/or it can possess undesirable effects. Later on, in the drug discovery and development pipeline, hits might be developed into lead compounds that are subjected to optimization steps before achieving a preclinical agent. Unless a hit compound is discovered, it is hard to proceed in this drug discovery pipeline. Such a hit compound might be discovered through a variety of strategies that might utilize high-throughput screening, fragment-based methods, focused libraries, or repurposing molecules, as well as others. In lieu, it might be desirable to identify some hit TMPRSS2 inhibitor compounds to be advanced later in the drug discovery pipeline of new antiviral agents against coronaviruses.

Repurposing is one the drug discovery strategies that has been successfully applied [39,40,41,42]. Briefly, in such a strategy, compounds or drugs that have been studied for a certain therapeutic use would be re-investigated for a different therapeutic purpose. Adopting this method, repurposing of the anticoagulant drug nafamostat (**1**, Figure 1) and the anti-acute pancreatitis camostat (**2**, Figure 1) showed that they can potentially inhibit TMPRSS2-dependent viral entry, and they underwent clinical trials as possible COVID-19 treatments [15,17,43,44,45]. Nevertheless, they suffered drawbacks and limitations that stalled further advancement. Both nafamostat (**1**) and camostat (**2**) are irreversible covalent TMPRSS2 inhibitors where the ester moiety undergoes nucleophilic attack by Ser441 amino acid after the initial binding step of the drug within the binding site. While the ester moiety enables this second step of establishing a covalent bond with the TMPRSS2, it is also responsible for metabolic liability of nafamostat (**1**) and camostat (**2**) proved by their short plasma half-life of less than 1 and 23.1 min, respectively [15,46,47]. In addition, covalent inhibitors are notorious for high risk and toxicities because of off-target interactions [48]. Accordingly, rapid metabolic deactivation and off-target inhibition consequences could be major obstacles for development of covalent TMPRSS2 inhibitors despite their potent activity. Despite less potency, it might be desirable to identify reversible TMPRSS2 inhibitor hit compounds that would possibly open a gateway for development of a common treatment for SARS-CoV-2, SARS-CoV, MERS-CoV, and influenza A viruses.

Analysis of structural features of nafamostat (**1**) and camostat (**2**) shows that, in addition to the common central ester group, they contain a common left 4-guanidinophenyl moiety that might play a critical role in binding to the target protein [43,49,50]. Referring to X-ray crystal structure (PDB ID: 7MEQ), this 4-guanidinophenyl moiety is found covalently bound as a result of nucleophilic attack on ester moiety by the Ser441 amino acid of TMPRSS2. While nafamostat (**1**) possesses an amidinonaphthyl fragment on the right of the central ester moiety, camostat (**2**) has a phenyl moiety bearing a longer aliphatic substituent. Despite this right moiety being a leaving moiety that is absent in the X-ray crustal structure, it might affect the first step of ligand binding prior to the subsequent step of establishing the covalent binding. This might be supported by the found higher potency of nafamostat relative to camostat [51]. Guanidine and amidine are two closely related functional groups that differ by only one nitrogen atom. In literature, several amide-based peptidomimetics incorporating amidine or guanidine functionalities were reported as TMPRSS2 inhibitors, such as 3-amidinophenylalanyl-derived inhibitor **3** [52,53] and ketobenzothiazole-containing diguanidine inhibitor **4** [54,55]. However, it is known that amide-based peptidomimetics, despite being potent inhibitors, have disappointing activity because of their metabolic liabilities and poor pharmacokinetics [56,57]. In this regard, the small molecule non-peptidomimetic inhibitors (AKA class C and D mimetics) offer more advantages [58,59,60,61,62]. Interestingly, it was found that the small molecules pentamidine (**5**) and propamidine (**6**), in which the right amidinoaryl feature was conserved in the form of 4-amidinophenyl moiety and the left 4-guanidinophenyl moiety was converted also into 4-amidinophenyl moiety while the covalent binding-responsible and metabolically labile central ester moiety was replaced by flexible alkyl chains of variable lengths, were found to possess TMPRSS2 inhibitory activity [50,51].

Deployment of in silico methods for drug discovery and design prior to synthesis and biological evaluation enables efficient utilization of limited resources as it helps to eliminate the likely-to-fail ligands and directs synthetic and biological evaluation efforts towards possibly promising compounds. In this regard, it might be helpful to establish a small focused library of compounds based on available data for in silico investigation prior to synthesis and in vitro testing [37]. Considering the structural features of nafamostat (**1**), camostat (**2**), amidinophenylalanyl-derived inhibitor **3**, and ketobenzothiazole-containing diguanidine inhibitor **4**, coupled with those of pentamidine (**5**) and propamidine (**6**), a small, focused library of eight compounds (**7**–**14**, Figure 1) was designed for in silico study prior to synthesis and evaluation. While the covalent binding-responsible and metabolically labile central ester moiety no longer exists in pentamidine (**5**) and propamidine (**6**), the flexibility of their central alkyl chain translates into probability of multiple conformers. In fact, the desired biological activity might be associated with some conformers while other conformers do not contribute to the desired bioactivity and, even worse, might trigger undesirable effects and/or toxicity. Therefore, rigidification of flexible moieties might be helpful if the new structure is conformationally locked in a configuration that mimics the conformer mediating the desired activity. Accordingly, the six-membered phenyl or pyridinyl as well as the five-membered furanyl aromatic moieties were introduced as rigid central moieties replacing the flexible alkyl chains of pentamidine (**5**) and propamidine (**6**). As biological evaluation showed that the size of the central moiety impacts the activity of pentamidine (**5**) and propamidine (**6**) where pentamidine turned out to be more active than propamidine [51], compounds incorporating the larger three rings-based rigid system 2,6-bisphenylpyridine, were considered as a central moiety to investigate the size impact. In addition to unsubstituted amidines, the effect of substituent introduction as well as incorporation of the amidine moiety into a cyclic moiety such as tetrahydropyrimidine were also considered. Furthermore, the aromatic ring bearing the amidine group in the terminal right and left moieties varied between the monocyclic phenyl moiety and the heterobicyclic benzimidazolyl moiety. A literature search showed that most of these postulated structurally diverse compounds were previously synthesized and explored as antiparasitic agents [28,29,63,64]. Accordingly, their re-investigation as TMPRSS2 inhibitors might be considered as a repurposing effort. The established small library was subjected to in silico study followed by in vitro evaluations to confirm their activity as possible hit compounds for development of therapeutics against coronaviruses diseases.

### 3.2. In Silico Evaluation

Prior to preparation and evaluation of members of the established small library, compounds were subjected to in silico evaluation to advance only the structures predicted to possess the desired bioactivity, and eliminate those likely not active. Accordingly, a molecular docking study was conducted using the reported crystal structure of TMPRSS2 (PDB ID: 7MEQ) and the docked poses and binding interactions of these compounds within the substrate binding site of TMPRSS2 were examined. The reported crystal structure of TMPRSS2 showed that the left 4-guanidinobenzoyl fragment of nafamostat was covalently bound through the hydrolyzed ester moiety to Ser441, which is one of the TMPRSS2 catalytic triad residues (Asp435, Ser441, and His296). This fragment of nafamostat establishes a network of favorable interactions that involve the hydrogen bond between its carbonyl and Gln438 and Gly439 residues in addition to hydrogen bonding interactions between the guanidine functionality with Asp435, Ser436, Gly464 and Gly472 residues. In fact, the presence of a guanidinium moiety in a structure of proposed inhibitor was found to be crucial to the binding with TMPRSS2 [17,65].

While the investigated eight compounds could dock into the substrate binding site (Figure 2), binding scores of compounds **7**–**9** possessing the five-membered furan ring as central moiety coupled with phenyl moiety as the aromatic ring bearing the amidine group in the terminal right and left moieties were, in general, relatively lower than compounds **10**–**14** possessing the six-membered phenyl or pyridine containing central moieties coupled with benzimidazole as the aromatic ring bearing the amidine group in the terminal right and left moieties (Table 1). Amongst furan-based compounds **7**–**9**, only compound **7** having unsubstituted amidines could establish an interaction with one of the catalytic triad amino acid residues (Asp435). Meanwhile, compounds **8** and **9**, in which the amidine is substituted or incorporated within a cycle, despite a higher binding score relative to compound **7**, could not establish any interaction with the catalytic triad. Interestingly, in silico results showed that compounds **10**–**14**, possessing the six-membered phenyl or pyridine containing central moieties and moieties coupled with benzimidazole as the aromatic ring bearing the amidine groups, could establish one or more favorable interactions with the catalytic triad amino acid residues (Table 1). Binding scores of compounds **11** and **14**, possessing isopropyl-substituted amidine moieties, were higher than corresponding compounds **10** and **13** possessing unsubstituted amidine moieties. Furthermore, compounds **13** and **14** with the larger three rings-based system 2,6-bisphenylpyridine central moiety showed higher binding scores relative to corresponding compounds **10** and **11** having the smaller pyridine ring as central moiety. As the catalytic triad residues (Asp435, Ser441, and His296) are crucial for activity of TMPRSS2, a predicted interaction with at least one of these residues would translate into a significant inhibitory activity, and the established interactions by compounds **7** and **10**–**14** with triad residues and other amino acids were further scrutinized. The catalytic Asp435 was the most frequent triad residue involved in favorable interactions with compounds **7** and **10**–**14**; specifically, with the amidine moieties. It is noteworthy to mention that such interaction with Asp435 was amongst the potential interactions detected in the X-ray crystal structure of TMPRSS2 with the nafamostat fragment (PDB ID: 7MEQ). However, interaction with Asp435 was not predicted for compound **14**, having the highest docking score, but instead interactions with the other two catalytic triad residues Ser441 and His296 were predicted. Favorable interactions with Ser441 were also predicted for compounds **12** and **13** while favorable interaction with His296 was the only triad interaction predicted for compound **11**. Meanwhile, Asp435 was the sole triad residue involved in two favorable interactions with compounds **10** and one favorable interaction with compound **7**. In addition to interactions with triad residues, favorable interactions of the amidine moiety with Ser436 (for compounds **7**, **12**, **13**, and **14**) and Gly464 (for compounds **7** and **10**) were predicted. Such interactions with Ser436 and Gly464 were amongst the potential detected interactions for the guanidine moiety of nafamostat fragment in the X-ray crystal structure (PDB ID: 7MEQ).

Based on these results, compounds **8** and **9**, which missed all interactions with the catalytic triad residue were excluded from further biological evaluation. As compounds **7** and **10**–**14** were predicted to establish a more or less diverse interaction pattern with one or more catalytic triad residues (one Asp435 interaction for compound **7**, two Asp435 interaction for compound **10**, Asp435 and Ser441 interactions for compounds **12** and **13**, Ser441 and His296 interactions for compound **14**, and one His296 interaction for compound **11**) coupled with calculated favorable binding scores, they were advanced to biological evaluation to assess their inhibitory activity on TMPRSS2.

### 3.3. Chemistry

Synthesis of compounds **7**, **13**, and **14** was reported previously [28,29]. Synthesis of compounds **10**–**12** is outlined in Figure 1. 4-Amidino-1,2-phenylenediamine derivatives **16** were prepared from commercially available 3,4-diaminobenzonitrile by applying Pinner reaction conditions [66]. Bis-Aldehydes **15** were reacted with 4-amidino-1,2-phenylenediamine derivatives **16** in ethanol in the presence of sodium metabisulphite to furnish the crude final compounds [67]. After sodium hydroxide treatment, crude compounds were converted to the hydrochloride salt by stirring with ethanolic HCl to furnish the desired compounds **10**–**12** as HCl salts.

### 3.4. In Vitro Evaluations

#### 3.4.1. TMPRSS2 Inhibition Assay

Based on in silico calculations, compounds **7** and **10**–**14** that could establish interactions with one or more of the catalytic triad residues were advanced for in vitro evaluation to assess their inhibitory activity on TMPRSS2. For comparison, pentamidine and nafamostat were used as reference standard compounds. While nafamostat, the covalent irreversible inhibitor showed less than 1 nM IC_50_ value, pentamidine exhibited nearly 40 µM IC_50_. Interestingly, the low micromolar IC_50_ values of compounds **12** and **10** were comparable to the reference pentamidine (Table 2), suggesting their potential inhibitory activity against TMPRSS2. Structurally, both compounds **12** and **10** have 6-amidinobenzimidazol-2-yl moieties bearing no substituents on the amidine fragment, but compound **12** has the six-membered phenyl moiety as the molecule’s central moiety instead of the six-membered pyridine moiety of compound **10**. In comparison, compound **11** having the same structure as compound **10** but having *N*-isopropyl substituents at the amidine moieties, possessed a considerably lower potency relative to compound **10**. The results showed that increasing the size of the central moiety of compounds **10** and **11** from the one ring-based pyridine ring to the three rings-based 2,6-bisphenylpyridine system in compounds **13** and **14** resulted in a high decrease in TMPRSS2 inhibitory activity. Thus, compounds **13** and **14** showed high micromolar IC_50_ values (Table 2). Comparing results of compounds **13** and **14** considering their structures, it can be inferred that introduction of *N*-isopropyl substituents at the amidine moieties (compound **14**) resulted in lowering the potency relative to the *N*-unsubstituted structure (compound **13**). Similarly, compound **7** having the smaller 4-amidinophenyl moiety as left and right moieties coupled with the five-membered furan ting as a central moiety possessed a low potency reflected by the found high micromolar IC_50_ value (Table 2).

#### 3.4.2. Cell Fusion Assays

To assess the inhibitory effect of compounds **7** and **10**–**14**, the previously developed SARS-CoV-2 S protein/TMPRSS2/ACE2 assay (S2TA assay) for quantitation of the impact on ACE2/TMPRSS2-dependant SARS-CoV-2 S protein-mediated membrane fusion utilizing “Dual Split Protein (DSP)” reporter complex [31] was used (Appendix A). As used target cells encompass DSP1-7 while effector cells encompass DSP8-11, fusion of effector cells with target cells will bring DSP together, resulting in reassociation to form functional Renilla luciferase (RL) and green fluorescent protein (GFP), which can be detected by luminescence or fluorescence. As tested compounds might elicit TMPRSS2-independent effects on membrane fusion, or affects reassociation of DSP proteins and, thus, the measured luminescence or fluorescence signals, an assay employing cells co-transfected with DSP1-7 and DSP8-11 was employed (CoTF assay) to account for the inhibition of membrane fusion mediated by targets other than TMPRSS2. In addition to compounds **7** and **10**–**14**, pentamidine and nafamostat were used as reference standard compounds. As compounds **7**, **10**–**14** and pentamidine exhibited micromolar IC_50_ values, their percent inhibition values were first assessed at 100 µM concentration. The results are summarized in Table 2.

The reference compounds pentamidine and nafamostat triggered an inhibition percent of 71.30 and 89.32%, respectively, at 100 µM concentration, and an IC_50_ value of 32.27 µM and 7.1 nM, respectively, in S2TA assay. It was interesting to find that compounds **10**, **13** and **14** triggered high inhibition percentages of 89.53, 93.98, and 97.33%, respectively, which were relatively higher than pentamidine, Table 2. However, the TMPRSS2 enzymatic assay showed that compound **10** exhibited a comparable IC_50_ value to that of pentamidine, while compounds **13** and **14** were impotent TMPRSS2 inhibitors. Since the employed DSP assay evaluated the inhibitory effect not only on TMPRSS2 but also on various mechanisms required for membrane fusion of SARS-CoV-2 in cells, it might be possible that compounds **13** and **14** inhibited membrane fusion through other mechanisms. Noteworthy, it was reported that multiple membrane serine proteases can replace TMPRSS2 for membrane fusion [68]. Nevertheless, DSP assay can be used to evaluate the effects of compounds in cellular assay with high throughput and without the use of infectious viral particles, and various SARS-CoV-2 inhibitors have been found using this assay [30,31,69]. Although compound **10** has the one ring-based central moiety while compounds **13** and **14** have the larger three rings-based central moiety, all of them share the presence of a pyridine ring in the central moiety. Despite predicted close in silico binding scores and the measured close IC_50_ value for TMPRSS2 inhibition, compound **12** possessing phenyl-based central moiety showed modest inhibition percent in DSP assay compared with the corresponding compound **10** possessing pyridine-based central moiety that showed potential activity (Table 2). This result reinforces the inferred contribution of mechanisms other than TMPRSS2 to the outcome of the DSP assay. In the case of compound **7** possessing the five-membered furan ring as the central moiety which was an impotent TMPRSS2 inhibitor (Table 2), it showed a low inhibition percent of 39.91%. Meanwhile, compound **11** which differed from compound **10** only by having isopropyl-substituted amidine moieties and possessed lower TMPRSS2 potency relative to compound **10**, was revealed to elicit a low inhibitory activity of 44.17% (Table 2). While IC_50_ values of low ineffective compounds **7**, **11** and **12** were above 100 µM, compounds **10**, **13** and **14** which triggered high inhibition percentages elicited better IC_50_ values than the 32.27 µM IC_50_ value of the reference reversible TMPRSS2 inhibitor pentamidine. Thus, compounds **10**, **13** and **14** possessed the more potent IC_50_ values of 10.87, 13.70, and 16.93, respectively. Next, evaluation of compounds **7** and **10**–**14**, pentamidine and nafamostat in CoTF assay, revealed that all of them had more than 100 µM IC_50_ values except for compounds **14**, **13** and **10** that showed IC_50_ values of 30.97, 37.00 and 37.84 µM, respectively.

#### 3.4.3. Pseudovirus Entry and Cell Viability Assays

To evaluate the capability of the tested compounds to inhibit viral entry into cells, SARS-CoV-2 pseudovirus infection was assessed in the presence of compounds **7** and **10**–**14**, pentamidine and nafamostat. Initially, their percent inhibition values were first assessed at 100 µM concentration before assessing their IC_50_ values. Cellular viability was assessed to confirm that inhibition results were not because of cell death. The results are summarized in Table 3.

As shown in Table 3, neither the tested compounds, pentamidine or nafamostat showed potential cytotoxic activity, as IC_50_ values for inhibition of cellular viabilities were more than 100 µM. While the standard pentamidine showed 69.48% inhibition of pseudovirus infection at 100 µM concentration and an IC_50_ value of 45 µM, only compound **14** showed a considerable 77.44% inhibition at 100 µM concentration and IC_50_ value of 83.66 µM. All other tested compounds possessed low percentage inhibition values of less than 21% at the 100 µM concentration. As compound **14** had a disappointing >100 IC_50_ value for TMPRSS2 in enzymatic assay but considerable activities in cellular DSP and pseudovirus entry assays, it might be inferred that its inhibitory activity could be mediated through molecular targets other than TMPRSS2. Meanwhile, the low micromolar TMPRSS2 inhibitory activity found for compounds **10**–**12** at the enzymatic level is not sufficient to trigger considerable activity at the cellular level. However, these compounds might serve as hit starting point compounds for development of more potential inhibitors.

#### 3.4.4. Thrombin and Factor Xa Enzyme Inhibition Assays

Coagulopathy is a serious complication of coronavirus infections including SARS-CoV-2, SARS-CoV-1, MERS-CoV and others [70,71]. Formation of blood clots might result in thromboembolism that could be life-threatening. In this regard, a compound that inhibits thrombin or factor Xa might be double-edged. It might minimize the risk of thromboembolism, yet it might also be associated with a bleeding risk. As for the overlap of TMPRSS2, thrombin and factor Xa substrates [72], compounds **7** and **10**–**14**, pentamidine and nafamostat were subjected to enzymatic assays for the inhibition of thrombin and factor Xa to assess their possible inhibitory effects. The outcome is summarized in Table 4.

While nafamostat inhibited thrombin with a nanomolar IC_50_ value of 34.1 nM, it also inhibited factor Xa but with a low micromolar IC_50_ value of 1.74 µM. Meanwhile, pentamidine might be relatively less capable of triggering bleeding while still inhibiting thrombin and factor Xa with a low micromolar IC_50_ value of 1.51 and 6.22 µM, respectively. Amongst evaluated compounds, compound **14**, which was the most effective inhibitor of viral infection despite it not being a potential TMPRSS2 inhibitor, possessed comparable IC_50_ values to those of pentamidine for the inhibition of thrombin and factor Xa (Table 4). Compound **13** had greater potential to inhibit thrombin and factor Xa relative to compound **14** but was still comparable to pentamidine. Structurally, compound **13** was relevant to compound **14** but had unsubstituted amidine moieties. Meanwhile, compounds **10**, **11** and **12**, which had the most potential amongst tested compounds to inhibit TMPRSS2, showed submicromolar to low micromolar IC_50_ values for thrombin inhibition which were still comparable to pentamidine. However, compounds **10** and **11**, having pyridine ring as a central moiety, were greater potential inhibitors of factor Xa relative to compound **13** having the central phenyl ring instead of the pyridine. In comparison with compounds **10**–**14**, compound **7**, having the furan ring as a central moiety with the smaller 4-amidinophenyl moiety as left and right moieties, lacked potential thrombin inhibitory activity and also had 5-fold less potential factor Xa inhibitor relative to pentamidine (Table 4).

## 4. Discussion

Coronaviruses are notorious for triggering outbreaks of epidemic viral infections which are characterized by being contagious and morbid such as MERS-CoV, SARS-CoV, and SARS-CoV-2. Despite the attempted development of vaccines and therapeutics targeting viral proteins, continuous viral mutation raises major issues that challenge such efforts. As coronaviruses require a conserved host-based mechanism for viral entry, targeting this mechanism rather than other viral-dependent mechanisms might offer common and effective tools against coronaviruses. TMPRSS2 is involved in this process of host-dependent mechanism of viral entry into the lung epithelial cell in vitro [73] and in vivo [74]. While nafamostat and camostat were discovered as potential TMPRSS2 inhibitors, they are metabolically labile, covalent inhibitors and, furthermore, failed clinical studies [75,76,77]. To circumvent limitations of covalent inhibitors and metabolic liability of nafamostat and camostat, development of a reversible TMPRSS2 inhibitor lacking the central ester group responsible for metabolic instability and irreversible covalent inhibition might be a promising strategy despite the anticipated lower potency. Considering similarity of the structural features of pentamidine with nafamostat, camostat, ketobenzothiazole-containing and amidinophenylalanine-derived peptidomimetic inhibitors, pentamidine might offer a suitable starting point to develop potential TMPRSS2 inhibitors. However, the central alkyl chain of pentamidine renders it a flexible molecule with multiple conformers. Structure rigidification via replacement of the flexible moieties by cyclic fragments would result in a beneficial conformational lock. To identify new hit molecules, a small set of eight compounds (**7**–**14**) were designed with diverse structural features considering size of the introduced cyclic central moiety and the terminal aryl moiety bearing the amidine moiety in addition to the absence/presence of substituents at the amidine moiety or incorporation into a ring. Considering that the utility of conducting in silico studies in initial steps to guide the selection of structures more likely to possess desired activity and to reduce the workload and discovery costs, in silico study was conducted for compounds (**7**–**14**) using the reported crystal structure of TMPRSS2 (PDB ID: 7MEQ). In silico results predicted that compounds **7**–**9** incorporating the five-membered furan ring as the central moiety were less able to establish interactions with residues of the catalytic triad than compounds **10**–**14** possessing central moieties incorporating a six-membered ring. Meanwhile, compounds **14** and **13** were also able to establish interactions with catalytic triad residues with calculated good binding scores. Based on interactions and binding scores, in silico study enabled the filtering-out of compounds **8** and **9** from further consideration. Consequently, compounds **7** and **10**–**14** were advanced for preparation and in vitro evaluation. The found in vitro low TMPRSS2 inhibitory activity for TMPRSS2 inhibition by compound **7** possessing the five-membered furan ring-based central moiety was in agreement with its predicted in silico low binding score. In vitro assay for TMPRSS2 inhibition confirmed the influential role for the size of the central moiety as compounds **10**–**12**, possessing one six-membered-based central moieties, had potential low micromolar activity comparable to pentamidine, while compounds **14** and **13**, possessing the larger three rings-based rigid system 2,6-bisphenylpyridine as central moiety, showed low TMPRSS2 inhibitory activity. Meanwhile, the presence of *N*-isopropyl substituents at the amidine moieties had a negative influence on TMPRSS2 inhibitory activity. As cellular activity is very important, a known experimental model from the literature for quantification of the inhibitory activity of a compound on membrane fusion and cell entry employed using DSP was addressed [31]. Surprisingly, compounds **14**, **13**, despite not being potential TMPRSS2 inhibitors, were the most active compounds triggering high inhibition percentages and low micromolar IC_50_ values in the conducted cell-based assay. Meanwhile, compound **10**, possessing potentially low micromolar IC_50_ for TMPRSS2 inhibition in enzymatic assay was less active, and compounds **11** and **12**, possessing also potential TMPRSS2 inhibitory activity were of low cellular activity. Considering that this assay evaluated the inhibitory effect not only on TMPRSS2 but also on various mechanisms involved for membrane fusion of SARS-CoV-2 in cells, coupled with the reported finding that multiple membrane serine proteases can replace TMPRSS2 for membrane fusion [68], it might be inferred that compounds **14** and **13** might act by mechanisms other than TMPRSS2 inhibition. These findings emphasize the importance of evaluating the effects of compounds in cellular assay as it can help to identify potential compounds with different molecular targets. Structure analysis considering these results revealed that compounds **14**, **13**, and **11**, sharing the presence of the six-membered pyridine ring in their central moiety, and **14** and **13**, having the larger 2,6-bis(phenyl)pyridine moiety, were the most active. Meanwhile, phenyl-based central moiety afforded the less potent compound **12**. To check for activity that might arise from inhibition of molecular targets other than TMPRSS2, as well as checking for reduction in measured luminescence and fluorescence that might arise from the effect on DSP reassociation, a CoTF assay was performed. Compounds **14**, **13** and **10** showed IC_50_ values of 30.97, 37.00 and 37.84 µM in the CoTF assay, respectively. IC_50_ values for the CoTF assay were higher than the IC_50_ values for the TMPRSS2-dependent membrane fusion assay (S2TA assay), indicating that these compounds have inhibitory effects on membrane fusion, but their inhibitory activity is possibly accompanied by TMPRSS2-independent inhibition of membrane fusion or effects on reassociation of reporter DSP proteins. Therefore, the compounds’ effects on pseudovirus entry and cell viability assays were assessed. All tested compounds showed no potential cytotoxic activity. The results showed again that compound **14**, lacking potential TMPRSS2 inhibition, was the most active inhibitor for pseudovirus entry amongst the tested compounds, while other compounds including the most effective TMPRSS2 inhibitors **10**–**12** exhibited low inhibitory activity for pseudovirus entry. This reinforces the conclusion that compound **14** has an inhibitory activity on viral entry not associated with TMPRSS2 inhibition. As coagulopathy and formation of blood clots is a serious complication of coronaviruses infections, inhibition of thrombin and factor Xa might have some benefits. However, inhibition of thrombin and factor Xa also bears bleeding risks. Consequently, thrombin and factor Xa inhibition by tested compounds was checked. Except for compound **7** that did not show potential inhibition of thrombin and factor Xa, all other tested compounds triggered thrombin and factor Xa inhibition comparable to the drug pentamidine. Similar to the systemically-used drug pentamidine, monitoring bleeding risks should be considered upon the use of this class of compounds. In conclusion, compound **14** might serve as a hit compound that might require further development into lead compounds against cellular entry of coronaviruses.

## Data Availability

No data were used in this study.

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
