# Peer review of "In Silico and In Vitro Evaluation of Some Amidine Derivatives as Hit Compounds towards Development of Inhibitors against Coronavirus Diseases"

_viruses, 2023, doi:10.3390/v15051171_

Round 1

Reviewer 1 Report

The manuscript focuses on very important pharmacological target, TMPRSS2 and its involvement of the spread of coronavirus diseases. However, there are some sections which should be significantly improved.

1. I did not see any references in connection with other amidinophenylalanine-derived  and ketobenzothiazole-type TMPRSS2 inhibitors. It should be incorporated not only in Introduction but it also should be presented/interpreted in Discussion part and compare them with the presenetd inhibitors (advantages, disadvantages, structural similarity etc)..

 Meyer, D.; Sielaff, F.; Hammami, M.; Böttcher-Friebertshäuser, E.; Garten, W.; Steinmetzer, T. Identification of the first synthetic inhibitors of the type II transmembrane serine protease TMPRSS2 suitable for inhibition of influenza virus activation. Biochem J. 2013, 452(2), 331–343. https://doi.org/10.1042/bj20130101

  Pilgram, O.; Keils, A.; Benary, G.E.; Müller, J.; Merkl, S.; Ngaha, S.; Huber, S.; Chevillard, F.; Harbig, A.; Magdolen, V.; Heine, A.; Böttcher-Friebertshäuser, E.; Steinmetzer, T. Improving the selectivity of 3-amidinophenylalanine-derived matriptase inhibitors. Eur. J. Med. Chem. 2022, 238, 114437. doi: 10.1016/j.ejmech.2022.114437.

   Colombo, E.; Désilets, A.; Duchêne, D.; Chagnon, F.; Najmanovich, R.; Leduc, R.; Marsault, E. Design and synthesis of potent, selective inhibitors of matriptase. ACS Med. Chem. Lett. 2012, 3(7), 530-534. doi: 10.1021/ml3000534.

      Beaulieu, A.; Gravel, É.; Cloutier, A.; Marois, I.; Colombo, É.; Désilets, A.; Verreault, C.; Leduc, R.; Marsault, É.; Richter, M.V. Matriptase proteolytically activates influenza virus and promotes multicycle replication in the human airway epithelium. J Virol. 2013, 87(8), 4237-4251. doi: 10.1128/JVI.03005-12.

2.  S2TA assay and CoTF assays should be more detailed presented. For example What are the limitations for these assays? etc

3. Table There are ND rows. Please measure the data at least in case of nafamostat.

4. Camostate failed during clinical testing. Why it is not updated?

5. Why Ki/IC50 values were not determined when active form of TMPRSS2 enzyme available. These enzyme kinetic measurements have to be done. Several amidinophenylalanine- or -ketobenzothiazole-derived peptidomimetic inhibitors possess nanomolar Ki/IC50 for TMPRSS2. It should be explained why the reported compounds here would be better with their microM IC50s.

6. Because these compounds are not specific? (selective maybe), please include thrombin, FXa enzyme kinetic measurements to evaluate the bleeding risk.

Minor English check is advisable (such as in the title coronaviruses diseases)

Author Response

Responses to comments from Reviewer 1

Comment: The manuscript focuses on very important pharmacological target, TMPRSS2 and its involvement of the spread of coronavirus diseases. However, there are some sections which should be significantly improved.

Response: We would like to thank the respected reviewer for his opinion and his efforts reviewing and commenting our work. We sincerely appreciate his insightful comments that allowed us to revise, correct and enhance our manuscript.

Comment:1. I did not see any references in connection with other amidinophenylalanine-derived  and ketobenzothiazole-type TMPRSS2 inhibitors. It should be incorporated not only in Introduction but it also should be presented/interpreted in Discussion part and compare them with the presenetd inhibitors (advantages, disadvantages, structural similarity etc)..

Meyer, D.; Sielaff, F.; Hammami, M.; Böttcher-Friebertshäuser, E.; Garten, W.; Steinmetzer, T. Identification of the first synthetic inhibitors of the type II transmembrane serine protease TMPRSS2 suitable for inhibition of influenza virus activation. Biochem J. 2013, 452(2), 331–343. https://doi.org/10.1042/bj20130101

Pilgram, O.; Keils, A.; Benary, G.E.; Müller, J.; Merkl, S.; Ngaha, S.; Huber, S.; Chevillard, F.; Harbig, A.; Magdolen, V.; Heine, A.; Böttcher-Friebertshäuser, E.; Steinmetzer, T. Improving the selectivity of 3-amidinophenylalanine-derived matriptase inhibitors. Eur. J. Med. Chem. 2022, 238, 114437. doi: 10.1016/j.ejmech.2022.114437.

Colombo, E.; Désilets, A.; Duchêne, D.; Chagnon, F.; Najmanovich, R.; Leduc, R.; Marsault, E. Design and synthesis of potent, selective inhibitors of matriptase. ACS Med. Chem. Lett. 2012, 3(7), 530-534. doi: 10.1021/ml3000534.

Beaulieu, A.; Gravel, É.; Cloutier, A.; Marois, I.; Colombo, É.; Désilets, A.; Verreault, C.; Leduc, R.; Marsault, É.; Richter, M.V. Matriptase proteolytically activates influenza virus and promotes multicycle replication in the human airway epithelium. J Virol. 2013, 87(8), 4237-4251. doi: 10.1128/JVI.03005-12.

Response: We sincerely thank the respected reviewer for drawing our attention to this issue. We have discussed and included in revised Figure 1 of our manuscript both of amidinophenylalanine-derived and ketobenzothiazole-type TMPRSS2 inhibitors. We have also cited all these mentioned references in our revised manuscript.  

Comment: 2.  S2TA assay and CoTF assays should be more detailed presented. For example What are the limitations for these assays? Etc

Response: We sincerely thank the respected reviewer for drawing our attention to this issue. We have presented in materials and methods the details of S2TA assay and CoTF assays and we have also included a Figure in the supplementary information illustrating such detailed information. We have mentioned in our manuscript that these are cellular assays and thus the measured response might be not only due to inhibition of TMPRSS2 activity, but the outcome might involve also effects by targets other than TMPRSS2 as well as effects on reassociation of DSP proteins.

Comment: 3. Table There are ND rows. Please measure the data at least in case of nafamostat.

Response: We express our sincere apologies for the respected reviewer for this missing data. We have fixed this issue and presented the determined values which were found to be >100 µM.

Comment: 4. Camostate failed during clinical testing. Why it is not updated?

Response: We express our sincere thanks to the respected reviewer for this issue. We have updated our manuscript with this information and included reference # 76 and 77 in the revised manuscript that support this updated information.

Comment: 5. Why Ki/IC50 values were not determined when active form of TMPRSS2 enzyme available. These enzyme kinetic measurements have to be done. Several amidinophenylalanine- or -ketobenzothiazole-derived peptidomimetic inhibitors possess nanomolar Ki/IC50 for TMPRSS2. It should be explained why the reported compounds here would be better with their microM IC50s.

Response: We would like to express our sincere thanks to the respected reviewer for his helpful comment. We have performed enzymatic assay for our compounds and updated our manuscript with the outcome of this TMPRSS2 enzymatic assay (Revised Table 2 and Section 3.4.1. TMPRSS2 inhibition assay). We have indicated in our manuscript that peptidomimetic inhibitors suffer several limitations that might overcame with the development of non-peptidomimetic small molecule inhibitors. These limitations might include their disappointing activity in vivo despite their potent in vitro activity which is contributed by their metabolic liabilities and poor pharmacokinetics. That is why it might be more promising to develop non-peptidomimetic small molecule inhibitors.   

Comment: 6. Because these compounds are not specific? (selective maybe), please include thrombin, FXa enzyme kinetic measurements to evaluate the bleeding risk.

Response: We would like to express our sincere thanks to the respected reviewer for his insightful comment that enabled us to enhance our manuscript. We have conducted thrombin and Factor Xa enzyme inhibition assays. The results showed that our compounds, in general, can inhibit thrombin and Factor Xa. The outcome results are included and discussed in our revised manuscript (Table 4 and Section 3.4.4. Thrombin and factor Xa enzyme inhibition assays).

Comment: Minor English check is advisable (such as in the title coronaviruses diseases)

Response: We would like to express our sincere thanks to the respected reviewer, we have reviewed the English writing and did a few alterations.

Reviewer 2 Report

Authors report here the in-silico target specific compound design approach against SARS-COV-2 TMPRRSS2 protein to develop new therapeutics. The approach is highly desired in view of limited therapeutics, although this is still in the initial phase in the drug discovery pipeline. Before this is accepted, authors need to make major changes in the manuscript

Major Comments

1.      Methods for in-silico screening used refers to 7-year-old manuscript, I am sure lot of steps have changed for rational design approaches, need the detailed steps in the main text, if space is the restriction, place it in the supplementary section,

2.      For the in-vitro evaluations using virus culture and target validations, write the methods/results separately, its confusing to see IC50 method is missing, I assume the target validation should be expressed as EC50?

3.       What is  the rational for using 100uM concentrations of each test and STD compounds for calculating % inhibition?

4.      Also where is the cytotoxicity evaluations to make sure IC50 are not sue to toxicity of compounds to host cells rather to virus

5.      Also, the primary goal of this rational design of target specific compounds is to overcome the limitations like covalent binding and metabolic liability of STD compounds already available, there is no assay or data to suggest otherwise, at least for compound 8, 11 and 12.

6.      Under the conclusion, authors states that compounds acted non specifically in CoTF assay, what other pathways these compounds could be acting, if not cell based assay, can authors prove using in-silico docking to see if these compounds are interacting with other host receptors?

Minor Comments

1.      On line 103, add micro symbol beside L, it shows 50 L

Author Response

Responses to comments from Reviewer 2

Comment: Authors report here the in-silico target specific compound design approach against SARS-COV-2 TMPRRSS2 protein to develop new therapeutics. The approach is highly desired in view of limited therapeutics, although this is still in the initial phase in the drug discovery pipeline. Before this is accepted, authors need to make major changes in the manuscript

Response: We would like to thank the respected reviewer for his opinion and his efforts reviewing and commenting our work. We sincerely appreciate his insightful comments that allowed us to revise, correct and enhance our manuscript.

Comment: Major Comments

  1. Methods for in-silico screening used refers to 7-year-old manuscript, I am sure lot of steps have changed for rational design approaches, need the detailed steps in the main text, if space is the restriction, place it in the supplementary section,

Response: We sincerely apologize to the respected reviewer for such outdated reference. We replaced it in the revised manuscript with new references from recent publications (years 2023~2020). Please, let us draw the attention of the respected reviewer that we have placed details of in silico screening in the supplementary materials as instructed in the comment.

Comment: 2. For the in-vitro evaluations using virus culture and target validations, write the methods/results separately, its confusing to see IC50 method is missing, I assume the target validation should be expressed as EC50?

Response: We sincerely apologize to the respected reviewer for this issue in previous version of our manuscript. In the revised manuscript, we have rewritten materials and methods introducing assays details. We have included all IC50 values for compounds in all conducted evaluations.

Comment: 3. What is the rational for using 100uM concentrations of each test and STD compounds for calculating % inhibition?

Response: We would like to express our sincere thanks to the respected reviewer for his comment. At the current level of our discovery work, we wish to identify hit compounds for further development. Therefore, it was needed to check whether the compound has activity or not in in the cell-based assay at 100 µM concentration before determining its potency. Consequently, we checked % inhibition of each tested compound at 100 µM. We checked also the standard compounds at the same concentration for comparison.

Comment: 4. Also where is the cytotoxicity evaluations to make sure IC50 are not sue to toxicity of compounds to host cells rather to virus

Response: We would like to express our sincere thanks to the respected reviewer for his comment. In the revised manuscript, we have conducted cell viability assays. The results showed that all tested compounds and standards has no potential cytotoxicity where the determined IC50 values were more than100 µM.

Comment: 5. Also, the primary goal of this rational design of target specific compounds is to overcome the limitations like covalent binding and metabolic liability of STD compounds already available, there is no assay or data to suggest otherwise, at least for compound 8, 11 and 12.

Response: We would like to express our sincere thanks to the respected reviewer for his insightful comment. Please, let us draw the attention of the respected reviewer that metabolic liability and covalent binding is related to chemical structure. In case of nafamostat and camostat, they are due to the presence of the ester group that undergo hydrolysis and nucleophilic attack. In the designed compounds, scaffold hopping of the soft ester group through replacement by the aromatic rings, which cannot undergo hydrolysis or covalent binding, was performed. The identified molecules are hit compounds that will undergo further development, this issue might checked for promising compound that might be obtained in later stages of development.

Comment: 6. Under the conclusion, authors states that compounds acted non specifically in CoTF assay, what other pathways these compounds could be acting, if not cell based assay, can authors prove using in-silico docking to see if these compounds are interacting with other host receptors?

Response: We would like to thank the respected reviewer for his comment. Please let us clarify that the words “non-specific inhibition” was corrected in the manuscript as we recognized that this expression does not convey appropriately the meaning we intended. While “non-specific inhibition” might mean an inhibition because of cell death, we actually were trying to say that there could be a measured inhibition that might arise from inhibition of molecular targets other than TMPRSS2 or inhibition measured luminescence and fluorescence that might arise from effect on DSP reassociation. Accordingly, we corrected this term in the revised manuscript. In fact, we found in literature that “Several membrane-associated serine proteinases might synergize with or replace TMPRSS2 as cellular activator of SARS-CoV-2” (J Biol Chem. 2021 Jan-Jun; 296: 100135. doi: 10.1074/jbc.REV120.015980). Therefore, we assessed inhibitory activity of our compounds on other serine proteases including thrombin and factor Xa. Interestingly, we found that our compounds can inhibit these proteases also. We presented these results in the revised manuscript.

Comment: Minor Comments

  1. On line 103, add micro symbol beside L, it shows 50 L

Response: We sincerely apologize to the respected reviewer for this mistake. We have corrected it in our revised manuscript.

Reviewer 3 Report

  • Major comments: 

In this study, Ahmed H.E. Hassan et al. evaluated a small library of eight compounds of amidine derivatives in silico and in vitro as hit compounds towards developing TMPRSS2 inhibitors against coronavirus diseases. They identified three compounds (12, 11, and 8) as potential hits inhibiting TMPRSS2-mediated membrane fusion with low micromolar IC50 values. However, they found that the inhibitory effects of these three compounds were associated with non-specific effects.

The study is relevant to the field and well-organized.

  • General concept comments

Here are some considerations/suggestions for the study:

1.     The major concern is that the inhibitory effects of the three compounds (12, 11, and 8) could be non-specific. Further experiments are needed to identify the chemical groups that are responsible for this non-specific inhibition, or further optimization of these three compounds is needed.

2.     A pseudovirus entry inhibition assay should be used to cross-check the S2TA assay.

3.     Another concern is that the authors didn’t provide the cytotoxicity data of the 6 compounds evaluated in the study. The cytotoxicity data is needed.

4.     It seems like compound 10 could be further developed owing to low non-specific inhibition.

5.     The introduction section should also provide a review of current TMPRSS2 inhibitors and their efficacy data on coronavirus prevention and treatment.

  • Specific comments:

1)    Please illustrate or detail the S2TA assay and materials that were being used.

2)    Line 103, 50 L?

3)    A higher resolution of Figure 2 should be provided. In addition, what do the different colors mean in the right panel of each subfigure? I recommend using dash lines of a different color to represent different interactions and make an explanation in the figure legend.

4)    Line 305, typo of “binging scores”.

Author Response

Responses to comments from Reviewer #3

Comment: In this study, Ahmed H.E. Hassan et al. evaluated a small library of eight compounds of amidine derivatives in silico and in vitro as hit compounds towards developing TMPRSS2 inhibitors against coronavirus diseases. They identified three compounds (12, 11, and 8) as potential hits inhibiting TMPRSS2-mediated membrane fusion with low micromolar IC50 values. However, they found that the inhibitory effects of these three compounds were associated with non-specific effects.

The study is relevant to the field and well-organized.

Response: We would like to thank the respected reviewer for his opinion and his efforts reviewing and commenting our work. We sincerely appreciate his insightful comments that allowed us to revise, correct and enhance our manuscript.

Comment: Here are some considerations/suggestions for the study:

  1. The major concern is that the inhibitory effects of the three compounds (12, 11, and 8) could be non-specific. Further experiments are needed to identify the chemical groups that are responsible for this non-specific inhibition, or further optimization of these three compounds is needed.

Response: We would like to thank the respected reviewer for his comment. Please let us clarify that the words “non-specific inhibition” was corrected in the manuscript as we recognized that this expression does not convey appropriately the meaning we intended. While “non-specific inhibition” might mean an inhibition because of cell death, we actually were trying to say that there could be a measured inhibition that might arise from inhibition of molecular targets other than TMPRSS2 or inhibition measured luminescence and fluorescence that might arise from effect on DSP reassociation. Accordingly, we corrected this term in the revised manuscript. In fact, we found in literature that “Several membrane-associated serine proteinases might synergize with or replace TMPRSS2 as cellular activator of SARS-CoV-2” (J Biol Chem. 2021 Jan-Jun; 296: 100135. doi: 10.1074/jbc.REV120.015980). Therefore, we assessed inhibitory activity of our compounds on other serine proteases including thrombin and factor Xa. Interestingly, we found that our compounds can inhibit these proteases also. We presented these results in the revised manuscript.

Comment: 2. A pseudovirus entry inhibition assay should be used to cross-check the S2TA assay.

Response: We would like to express our sincere appreciation to the respected reviewer for his insightful comment. Following his instructions to us, we have performed pseudovirus entry inhibition assay. The results showed that compound 14 is the most active inhibitor amongst tested compounds. We have presented and discussed the results of pseudovirus entry inhibition assay in our revised manuscript.

Comment: 3. Another concern is that the authors didn’t provide the cytotoxicity data of the 6 compounds evaluated in the study. The cytotoxicity data is needed.

Response: We would like to express our sincere thanks to the respected reviewer for his comment. We followed his comment and performed cell viability assay. We found that all tested compounds and employed standards have no potential cytotoxic effects with measured IC50 values more than 100 µM.

Comment: 4. It seems like compound 10 could be further developed owing to low non-specific inhibition.

Response: We would like to express our sincere thanks to the respected reviewer for his comment. After revision of the manuscript (including renumbering of compounds), we have concluded that compound 14 might be a candidate for further development as it elicited potential activities in cellular assays including pseudovirus entry inhibition assay and inhibition of cell membrane fusion.

Comment: 5. The introduction section should also provide a review of current TMPRSS2 inhibitors and their efficacy data on coronavirus prevention and treatment.

Response: We would like to express our sincere thanks to the respected reviewer for his comment. We have revised our manuscript and, in figure 1 and section “3.1. Design rational”, we have included peptidomimetic amidinophenylalanyl-derived inhibitors and ketobenzothiazole-containing diguanidine inhibitors, in addition to small molecule inhibitors nafamostat and camostat. We hope our revised manuscript would be informative and satisfactory to the respected reviewer.

  • Specific comments:

Comment: 1. Please illustrate or detail the S2TA assay and materials that were being used.

Response: We would like to express our sincere thanks to the respected reviewer for his comment. We included “Supplementary Fig. 1” to illustrate S2TA assay and CoTF assay.

Comment: 2. Line 103, 50 L?

Response: We sincerely apologize to the respected reviewer for this typos mistake. We have corrected it and added the micro symbol beside L.

Comment: 3. A higher resolution of Figure 2 should be provided. In addition, what do the different colors mean in the right panel of each subfigure? I recommend using dash lines of a different color to represent different interactions and make an explanation in the figure legend.

Response: We would like to express our sincere thanks to the respected reviewer for his comment. We have replaced the figure with higher resolution figure and we added legend for interaction types showing different colors for different interaction types. We hope the new figure would be satisfactory for the respected reviewer.

Comment: 4. Line 305, typo of “binging scores”.

Response: We sincerely apologize to the respected reviewer for this typos mistake. We have corrected typos in the manuscript.

Round 2

Reviewer 2 Report

Manuscript has been improved to a great extent with suggested experiments are being incorporated.

Reviewer 3 Report

I think that the manuscript has been improved, and the authors have addressed most of my concerns.